# Development and Validation of a Multi-Locus PCR-HRM Method for Species Identification in *Mytilus* Genus with Food Authenticity Purposes

**DOI:** 10.3390/foods10081684

**Published:** 2021-07-21

**Authors:** Marianela Quintrel, Felipe Jilberto, Matías Sepúlveda, María Elisa Marín, David Véliz, Cristián Araneda, María Angélica Larraín

**Affiliations:** 1Food Quality Research Center, Universidad de Chile, Santiago 8380544, Chile; marianela.quintrel@ug.uchile.cl (M.Q.); fjilberto@ug.uchile.cl (F.J.); matiasepulveda@ug.uchile.cl (M.S.); dveliz@uchile.cl (D.V.); craraned@uchile.cl (C.A.); 2Master in Food Program, Mention in Management, Quality and Food Safety, Facultad de Ciencias Químicas y Farmacéuticas, Universidad de Chile, Santiago 8380544, Chile; 3Master in Biological Sciences Program, Facultad de Ciencias, Universidad de Chile, Santiago 7800003, Chile; 4Master in Food Sciences Program, Facultad de Ciencias Químicas y Farmacéuticas, Universidad de Chile, Santiago 8380544, Chile; 5Departamento de Ciencia de los Alimentos y Tecnología Química, Facultad de Ciencias Químicas y Farmacéuticas, Universidad de Chile, Santiago 8380544, Chile; memarin@uchile.cl; 6Departamento de Ciencias Ecológicas, Facultad de Ciencias, Universidad de Chile, Santiago 7800003, Chile; 7Departamento de Producción Animal, Facultad de Ciencias Agronómicas, Universidad de Chile, Santiago 8820808, Chile

**Keywords:** *Mytilus*, species identification, validation, high-resolution melting, PCR

## Abstract

DNA-based methods using informative markers such as single nucleotide polymorphism (SNPs) are suitable for reliable species identification (SI) needed to enforce compliance with seafood labelling regulations (EU No.1379/2013). We developed a panel of 10 highly informative SNPs to be genotyped by PCR-High resolution melting (HRM) for SI in the *Mytilus* genus through in silico and in vitro stages. Its fitness for purpose and concordance were assessed by an internal validation process and by the transference to a second laboratory. The method was applicable to identify *M. chilensis, M. edulis, M. galloprovincialis* and *M. trossulus* mussels, fresh, frozen and canned with brine, oil and scallop sauce, but not in preserves containing acetic acid (wine vinegar) and tomato sauce. False-positive and negative rates were zero. Sensitivity, expressed as limit of detection (LOD), ranged between 5 and 8 ng/μL. The method was robust against small variations in DNA quality, annealing time and temperature, primer concentration, reaction volume and HRM kit. Reference materials and 220 samples were tested in an inter-laboratory assay obtaining an “almost perfect agreement” (κ = 0.925, *p* < 0.001). In conclusion, the method was suitable for the intended use and to be applied in the seafood industry.

## 1. Introduction

Mussels from *Mytilus* genus represented 20% of the worldwide mollusks production in 2018 [1]. *Mytilus* genus is one of the most cultivated and marketed bivalve, widely appreciated as a tasty and nutritious source of protein. Mussel aquaculture is a relevant economic activity for many coastal communities [2,3]. The related commercial species, *Mytilus edulis* (Linnaeus, 1758), *Mytilus galloprovincialis* (Lamarck, 1819), *Mytilus chilensis* (Hupé, 1854) and *Mytilus trossulus* (Gould, 1850) are taxonomically recognized in the World Register of Marine Species [4] and the Integrated Taxonomic Information System [5] together with other *Mytilus* spp. Nowadays, the European regulation on the common organization of the markets in fishery and aquaculture products [6] requests the declaration of the commercial designation of the species and its scientific name in the label. Each participant country has its official list of commercial designations and scientific names for fishery and aquaculture products (https://ec.europa.eu/fisheries/cfp/market/consumer-information/names_en, accessed on 21 May 2021). This specific traceability requirement is aimed at confirming the authenticity of the products. Species substitution can result in an inexpensive product being labelled as high-priced. It can also affect food safety via unnoticed consumption of allergens due to undeclared species [7,8]. Seafood mislabeling is well documented throughout history [9,10]; it impacts not only food authenticity [11] but also allows the trade in the markets of endangered species or products from illegal, unreported and unregulated (IUU) fisheries, threatening wildlife [12], hampering conservation and negatively affecting consumers decisions [13]. Nowadays, the breadth and depth of mislabeling are coming into sharper focus, thanks to DNA-based species verification methods [14].

DNA testing is currently reported as the regulatory tool of choice for seafood species identification (SI) [15,16,17]. Several DNA analysis techniques rely on Polymerase Chain Reaction (PCR) [18]; however, the informative potential of the targeted genomic regions should be rigorously tested before its adoption in a standardized system. Recently, PCR-based methods have been widely used in seafood authentication [19] and harvesting location detection [20].

For *Mytilus* SI in either fresh or processed products, different molecular markers and techniques have been developed [21,22,23,24]. However, the most common DNA marker used for this purpose targets the polyphenolic adhesive protein gene, highly conserved within, but polymorphic among species [22,24,25,26,27,28]. Targeting this region, Jilberto et al. [29] developed a high-resolution melting (HRM) assay that allowed them to identify *M. chilensis, M. edulis* and *M. galloprovincialis* and their hybrids. This technique is simple and affordable to most molecular analytical laboratories.

SI based in a single locus (“mono-locus”) is relatively easy to perform, but this approach can produce conflicting results among markers and/or when individuals from hybrid zones are analyzed. Single nucleotide polymorphism (SNP) multi-locus panels have allowed for identifying *M. edulis, M. galloprovincialis, M. trossulus* [30,31], *M. chilensis*, *M. planulatus* and *M. platensis* [32,33,34]. However, genotyping a high number of SNPs may not be practical in routine analysis; therefore, it is attractive to develop reduced panels selecting highly informative SNPs for SI. The identification of the most informative loci can be performed by different criteria such as *F_ST_* outliers or minor allele frequency (MAF) [35,36].

Before the application of an analytical method in food analysis, its performance should be evaluated through compliance with quality criteria according to international guidelines [37,38,39]. Moreover, in the case of laboratories involved in import and export food testing or law enforcement, compliance with ISO/IEC Standard 17025 requirements is necessary [40]. The fit for purpose and the performance of a qualitative real-time PCR method can be first validated in-house, assessing the specificity, sensitivity, repeatability, reproducibility and practicability. Moreover, the transference of the method to a second laboratory can be performed [38].

To support the confidence of consumers, the food industry, business operators and regulators about seafood authenticity, and to avoid the mislabeling of mussels, it is essential to have available an affordable method whose results are internationally recognized.

In this study, we aim to develop a multi-locus SNPs method based on the PCR-HRM analysis, for the identification of species of the *Mytilus* genus (*M. chilensis*, *M. edulis*, *M. galloprovincialis* and *M. trossulus*), and to assess its fitness for purpose by an in-house validation process.

## 2. Materials and Methods

### 2.1. Method Development and SNPs Selection

To select highly informative SNPs for *Mytilus* SI (*M. chilensis*, *M. edulis*, *M. galloprovincialis* and *M. trossulus*) we analyzed two SNP datasets: (A) 49 SNP discovered by data mining from EST in *M. edulis*, *M. galloprovincialis* and *M. trossulus*, genotyped by Sequenom^®^ MassARRAY iPLEX [41,42]. These SNPs were genotyped in 311 individuals previously identified with the nuclear PCR-RFLP marker ME 15–16, as 189 *M. chilensis*, 27 *M. edulis*, 78 *M. galloprovincialis* and 17 *M. trossulus.* The second dataset consists of (B) 74 SNPs discovered by Araneda, 2016 [43] using RADseq in *M. chilensis*, along with 16 SNPs from the previous dataset (from Zbawicka et al. [41,42]) genotyped by the GT-seq method in *M. chilensis, M. edulis* and *M. galloprovincialis*. These SNPs were genotyped in 386 individuals previously identified with the nuclear PCR-RFLP marker ME 15–16, as 263 *M. chilensis*, 41 *M. edulis* and 82 *M. galloprovincialis*.

In each dataset, the SNPs were selected using two criteria: the *F_ST_* outlier and the MAF. The first criterion (*F_ST_* outlier) selects those SNPs showing *F_ST_* values above the upper limit of the confidence interval (95%) of the distribution of *F_ST_* obtained with the FDIST2 method [44] implemented in LOSITAN [45] and configured according to Araneda et al. [43]. The rationale of this selection criterion is based on the fact that loci showing *F_ST_* outliers values are candidates for having undergone a selection and adaptation processes during speciation revealing high levels of genetic differentiation, increasing resolution for assigning individuals to species [45]. The second criterion, called MAF_MAX_, selects loci that showed maximum differences in the MAF between the analyzed species, calculated by adding the absolute values of the differences between the MAFs for each pair of species. Those SNPs whose MAF_MAX_ values are over the 80th percentile in the MAF_MAX_ distribution were considered the most informative. Using this criterion, we select SNPs that showed a fixed allele in one species and the alternative allele present, either fixed or not, in the other species.

To select the highly informative SNPs for SI, four groups of SNPs were tested in each dataset: the first group contained the loci selected by the *F_ST_* outlier criteria, the second group have the loci selected by MAF_MAX_ criteria, the third group contains loci selected by the *F_ST_* outlier and the MAF_MAX_ criteria simultaneously. The fourth group contains SNPs selected either by the *F_ST_* outlier or MAF_MAX_ criteria. The performance of each SNP group was tested, counting the individuals correctly assigned to species based on their multi-locus genotypes. Each individual was assigned to species, using the Training, Holdout and leave-one-out (THL) procedure (Anderson, 2010) implemented in GeneClass2 software [46], with the Bayesian method of [47]. We considered an individual assigned to a species when the probability of assignment was upper to 95% [48]; otherwise, the individual was considered a hybrid.

### 2.2. Sample Collection

Raw samples from natural populations were analyzed to test in vitro the performance of the selected SNPs. Individuals of the target species, *M. chilensis, M. edulis, M. galloprovincialis* and *M. trossulus* (34 per species, n = 136), were collected in four sites from putatively allopatric populations, avoiding described hybrid zones. The species of these samples was confirmed by PCR-RFLP Me15-16 *Aci*I [49] and/or PCR-HRM PAPM (high resolution melting analysis on the polyphenolic adhesive protein gene of mussels) [29]. An individual was considered for further analysis when the two SI methods gave the same result. In addition, 20 individuals from each of the other mussel species commonly found in mussel products (*Aulacomya atra* and *Choromytilus chorus*) and species closely related to the *Mytilus* genus (*Perumytilus purpuratus*) were collected from natural populations (Table 1; Appendix A—Table A1). SI in these specimens was performed by morphological characters in the whole individual, before tissue sampling.

To determine the applicability of the method, canned and frozen pre-cooked commercial mussel products from Germany, Denmark, Spain and Chile were analyzed. Canned mussels with different packing media, “Brine”, “Oil”, “Marinated” and others with sauces (tomato, scallop and hot sauce), were included (Table 1). Of each can or package, three mussels (*n* = 3) were analyzed.

### 2.3. DNA Extraction

DNA was extracted from the mantle edge tissue of each individual by different methods: (i) phenol–chloroform method modified for *Mytilus* [50], (ii) cetyl trimethyl ammonium bromide (CTAB) modified by Arseneau et al. [51] and (iii) Chelex^®^ 100 method by Chai et al. [52]. The method (i) was used to extract individuals from natural populations and commercial samples; methods (ii) and (iii) were used to extract only the commercial samples. Methods (i) and (ii) include a RNAse treatment; however, as a rapid method, (iii) Chelex^®^ 100 does not. The quality and quantity of DNA were checked in 0.7% agarose gel in TBE buffer visualized with GelRed^®^ Nucleic Acid Gel Stain (Biotium Inc, San Francisco, CA, USA) and with a spectrophotometer NanoDrop 2000 (Thermo Fisher, Waltham, MA, USA) using only samples with absorbance ratio 260/280 nm > 1.8. All samples were diluted to a concentration of 20 ng/μL and stored at −18 °C for further molecular analysis.L and stored at −18 °C for further molecular analysis.

### 2.4. PCR-HRM Analysis and Primer Design

To genotype the selected informative SNP using PCR-HRM, primers were designed with Primer3 v4.1.0 [53] according to Jilberto et al. [29]. The PCR-HRM reaction was optimized at 96 °C for 2 min, followed by 35 cycles at 96 °C for 15 s and 59 °C for 30 s. The final melt curve analysis was performed from 55 °C to 95 °C, with increments of 0.1 °C per second. The reaction contained 0.1 mM of forward and reverse primers (Appendix A—Table A2) and 20 ng of genomic DNA and 1× of HRM SensiFast™ kit (Meridian Biosciences^®^, Cincinnati, OH, USA) in a final volume reaction of 8 μL. All analyses were performed in an ECO™ Real-Time PCR System (Illumina, Inc. San Diego, CA, USA) and a magnetic induction cycler quantitative PCR MIC qPCR Cycler (BioMolecular System, Brisbane Queensland, Australia) using the software provided with each equipment. The genotype of each locus was determined by comparing the shape of the melting curve of the query individual with the curve of a reference individual of each species. To be sure that the amplified fragment contains the target SNP, the amplicons of two individuals of each of the four target species were Sanger sequenced in Macrogen (Seoul, South Korea). For each SNP, the amplicon size and polymorphism position was confirmed.

### 2.5. Diagnostic Test Performance

We evaluated 216 individuals from natural populations (Table 1), calculating Sensitivity (S) and Specificity (E) as in Equation (1) and Equation (2) [54]:(1)$$S=\frac{N^{\underline{o}}\;\mathrm{individuals}\;\mathrm{with}\;\mathrm{the}\;\mathrm{species}\;\mathrm{correctly}\;\mathrm{identified}}{N^{\underline{o}}\;\mathrm{individuals}\;\mathrm{from}\;\mathrm{target}\;\mathrm{species}\;\mathrm{analyzed}}$$
(2)$$E=\frac{N^{\underline{o}}\;\mathrm{individuals}\;\mathrm{correctly}\;\mathrm{excluded}\;\mathrm{from}\;\mathrm{the}\;\mathrm{species}\;}{N^{\underline{o}}\;\mathrm{individuals}\;\mathrm{from}\;\mathrm{non}-\mathrm{target}\;\mathrm{species}\;\mathrm{analyzed}}$$

### 2.6. Fitness for Purpose Assessment of the Multi-Locus PCR-HRM Method for SI in Mytilus Genus

The qualitative method parameters evaluated in the in-house validation process were applicability, practicability, specificity of primers, sensitivity (limit of detection—LOD), robustness and method transferability [37,38].

#### 2.6.1. Applicability

The applicability or scope of application of the method [55] assessed the possibility of extracting DNA from frozen and canned commercial mussels that can be successfully analyzed by the SI HRM-PCR method (Table 1). Commercial samples were extracted by the phenol–chloroform, CTAB and Chelex methods, and the DNA quality was evaluated as described in Section 2.2. A DNA concentration ≥ 20 ng/µL was considered sufficient. The DNA that successfully met quality and concentration requirements was tested by the PCR-HRM multi-locus panel.

#### 2.6.2. Practicability

The assessment of the ease of operation and throughput of the method [55] considered the minimal unit of analysis, equipment availability, cost, length of the laboratory analysis, occupational safety and staff training requirements [38].

#### 2.6.3. Specificity of Primers

We evaluate the specificity of each primer pairs of the selected SNPs to ensure they amplify only in the target species. First, an in silico evaluation using Primer-BLAST was performed [56]. Secondly, we tested the specificity of each primer pair in vitro to detect any possible amplification in species close to the *Mytilus* genus (*Perumytilus purpuratus*) and other species commonly found in mussel products (*Aulacomya atra* and *Choromytilus chorus*). The false-negative rate (FNR) and the false-positive rate (FPR) were calculated as in Equations (3) and (4):(3)$$\;\mathrm{FNR}=\frac{\;N^{\underline{o}}\;\mathrm{individuals}\;\mathrm{from}\;\mathrm{known}\;\mathrm{target}\;\mathrm{species}\;\mathrm{misclassified}\;\mathrm{as}\;\mathrm{non}-\mathrm{target}\;\mathrm{species}\;}{\mathrm{Total}\;\mathrm{individuals}\;\mathrm{of}\;\mathrm{known}\;\mathrm{target}\;\mathrm{species}\;\mathrm{tested}}$$
(4)$$\mathrm{FPR}=\frac{\;N^{\underline{o}}\;\mathrm{individuals}\;\mathrm{from}\;\mathrm{known}\;\mathrm{non}-\mathrm{target}\;\mathrm{species}\;\mathrm{misclassified}\;\mathrm{as}\;\mathrm{target}\;\mathrm{species}\;}{\mathrm{Total}\;\mathrm{individuals}\;\mathrm{of}\;\mathrm{known}\;\mathrm{non}-\;\mathrm{target}\;\mathrm{species}\;\mathrm{tested}}$$

#### 2.6.4. Sensitivity (LOD)

To determine the limit of detection (LOD), the DNA concentration in ng/μL was expressed in Haploid Genome Equivalents (HGE) using the C-value. For *M. chilensis,* it was estimated as 1.56 pg, (an average among the species from *Mytilus* genus) and for the non-target species (*Aulacomya atra*), the C-value was 2.24 pg [57]. LOD correspond to the lowest concentration of the target species that can be detected by the method at least 95% of the time. A preliminary screening covering the range of 1000, 600, 200, 50, 10, 2 and 0.1 HGE was tested in the 10 SNPs. Following Broeders et al. [38] protocol, SNPs PAPM and L2 were analyzed in the DNA concentration range of 25, 20, 15, 12, 8, 5 and 1.25 ng/μL in 6-fold. The PCR-HRM was performed in the same conditions described in Section 2.3. The last dilution, where all six replicates gave a positive and specific amplification [58], together with higher and lower dilutions, were tested in 60 replicates [59] for SNP PAPM and L2. The lowest dilution level, at which all 60 replicates showed a specific positive amplification, was considered as the LOD [38].

#### 2.6.5. Robustness

Robustness was evaluated in triplicate for SNPs PAPM and L2 through the fractional factorial design described in Youden and Steiner (1975) [60]. Seven factors or experimental conditions in the protocol were varied to test how small, deliberate modifications affect the results. We considered the following factors: DNA quality through 260/230 and 260/280 ratios, annealing temperature and time, primers concentration, the total volume of reaction and two different HRM kits. Robustness was evaluated either in DNA extracted by the phenol–chloroform and CTAB methods. The standard value of the technique, along with the lowest and highest values for each experimental condition and its combination per run are shown in Table 2.

#### 2.6.6. Transferability Test

The standardized operational procedure (SOP) to perform the SI analysis and 220 blind samples, containing 44 of each of the target species (*M. chilensis, M. edulis, M. galloprovincialis*, *M. trossulus)* and 44 individuals of the non-target species (*Aulacomya atra* and *Choromytilus chorus*), were transferred to a second independent laboratory with different analysts and thermocyclers. The concordance of the results between both laboratories was analyzed using Cohen’s kappa coefficient (κ), where *p_o_* corresponds to the sum of observed concordances and *p_e_* to the sum of concordances attributable to chance (Equation (5)) [61].
(5)$$Κ=\frac{p_{o}-p_{e}}{1-p_{e}}\;$$

## 3. Results

### 3.1. Method Development and SNPs Selection

The group of SNPs that in silico analysis showed the best performance assigning individuals to species in dataset A were groups 2 (MAF_MAX_) and 4 (*F**_ST_* outliers or MAF_MAX_) (Table 3). Both panels correctly assigned 99.0% of the individuals to species, using only 17 and 22 SNPs, respectively. In dataset B, group 1 (*F**_ST_* outliers) showed the best performance with 27 SNPs, which correctly assigned 94.0% of the individuals to species. Consequently, we considered the 22 and 27 SNPs included in groups 4 and 1 in datasets A and B, respectively. Two SNPs were shared by both datasets, resulting in 47 most informative SNPs.

In these selected SNPs, all the sequences were checked in silico for any other polymorphism close to the SNP of interest. One SNP was removed because it can give unspecific curves during the HRM analysis. Another 14 were discarded to avoid SNP difficultly to genotype (A/T or C/G) or because it was not possible to design primers. From the remaining 32 SNPs, 12 we discarded because they gave redundant information (seven and five SNPs separate *M. trossulus* and *M. chilensis*, respectively, from the other species). Another three SNPs, having MAF values < 0.18 in one species and zero in the others, were considered not informative. Finally, two SNPs were discarded because no genotype was obtained for one of the species (Figure 1). On the other hand, one SNP A/T not considered by previous criteria was included because it was the only one whose allele frequencies were fixed and informative to separate *M. edulis* from *M. galloprovincialis.* After in silico selection, 16 SNPs were considered for the in vitro stage.

### 3.2. Sample Collection, DNA Extraction and Species Identification

The DNA obtained from raw individuals from natural populations satisfied the quality parameters for further analysis. In these samples, the species identified by mono-locus methods targeting the polyphenolic adhesive protein gene was confirmed by morphological traits. The results of the DNA extraction of twelve commercial samples of processed products are shown in Section 3.5.1 Applicability.

### 3.3. PCR-HRM Analysis and Primer Design

After the in vitro test, one SNP was removed because it amplified only in *Mytilus chilensis*. Another five SNPs were also discarded although they showed an observable amplification by qPCR, the obtained melting curves were not species-specific. Finally, the PCR-HRM method was successfully standardized in 10 of the most informative SNPs, named L2 to L10, also being the selected SNP amplified by PAPM primers from Jilberto et al. (2017), obtaining melting curves that allow to clearly distinguish each genotype (Figure 2).

Genotyping was successful for the 10 SNPs in all *Mytilus* individuals from natural populations. The amplicon sequencing of the 10 SNPs reveals that SNP loci were included in the amplified fragments. The 136 samples were correctly assigned to the expected species. The average assignment probability to each species was 100%.

### 3.4. Diagnostic Test Performance

All *Mytilus* individuals were correctly assigned to their species, and therefore all the individuals from other genus were correctly excluded. No false negative and false positive identifications were obtained; consequently, the specificity and sensitivity were maximum (S = 1.0, E = 1.0).

### 3.5. Fitness for Purpose Assessment of the Multi-Locus PCR-HRM Method for Species Identification in Mytilus Genus

#### 3.5.1. Applicability

Starting with a reduced DNA extraction trial, testing two individuals from only four of the canned commercial samples (brine, oil, marinated and with hot sauce as packing media), we concluded that the Chelex method yielded low-purity DNA, evidenced in a 260/280 ratio of 1.18 ± 0.05 (Appendix A—Table A3). Consequently, we continued testing only the phenol–chloroform and CTAB methods in the commercial products listed in Table 1. The average purity was similar between the phenol–chloroform (1.77 ± 0.08) and the CTAB (1.78 ± 0.26) methods and was within the acceptable range (1.7–2.0). However, the DNA purity from mussels canned with tomato sauce, extracted by the phenol–chloroform method, was below the accepted range (1.68 ± 0.07). Therefore, this method was excluded from subsequent HRM analysis.

The average DNA concentration was 771.29 ± 337.66 ng/μL for the phenol–chloroform method and 150.56 ± 89.63 ng/μL for the CTAB method, higher than the established quality limit (20 ng/μL) in all the tested products (Table 4).

The CTAB method gave 100% of specific HRM amplifications in canned mussels with brine and hot sauce, and in frozen pre-cooked commercial products. Consequently, it is only applicable to mussels packed in such matrices. Canned mussels in oil and scallop sauce gave 66.7% of successful HRM genotyping, but in mussels marinated and in tomato sauce, the genotyping success was lower than 50%.

An example of the results of SI in the commercial samples used for the applicability step is presented in Table 4. This table only shows samples with eight or more SNP successfully genotyped, allowing SI. Species were identified in frozen products and those packed in brine, oil, hot and scallop sauces. In products packed marinated or in tomatoes sauce, the lower number of SNPs genotyped hampered SI. Although canned marinated products are not within the scope of the method, it is interesting to note that the only species that could be identified in this matrix does not match the one declared in the label (Table 4).

#### 3.5.2. Practicability

Excluding the extraction step, the analysis time for a batch of seven samples (minimum unit of analysis) does not exceed 6 hours, considering one analyst. Real-time PCR equipment and HRM kits are usually available in molecular analysis laboratories. The cost of the necessary reagents to perform analysis per sample was estimated at USD 23. The reagents were not considered dangerous for the handler. The training time of an analyst is four hours. Consequently, the method is practical to apply under the described conditions.

#### 3.5.3. Specificity

In the in silico test, five primer pairs (L6–L10) showed no amplification with any sequence included in the NCBI database. Two primer pairs (PAPM and L2) showed amplification in at least one of the targets species and were considered species-specific. The remaining three pairs of primers (L3, L4 and L5) may not be specific because they showed amplification in non-target species.

During the in vitro test, two primer pairs amplify in non-target species: L7 in *C. chorus* and L9 in *A. atra* and *P. purpuratus.* The *Cq* values (PCR cycle number at which the reaction curve gets over the threshold line) were statistically similar (Appendix A—Table A4) to the target *Mytilus* species according to the non-parametric Gao test [62].

#### 3.5.4. Sensitivity (LOD)

In the preliminary screening, DNA samples of the target species *M. chilensis* ranged between 25 and 15 ng/μL and the 10 SNPs gave *Cq* values < 28. At these same concentrations, the non-target species *Aulacomya atra* gave *Cq* values > 30. This means that at the recommended DNA concentration (20 ng/μL), the method is sensitive, detecting the target species and excluding the non-target ones (Appendix A—Table A5). Considering the preliminary screening, when PAPM and L2 were genotyped in a narrower concentration range (25–1.25 ng/μL), the lowest dilutions at which all six repetitions gave a positive and specific amplification were 5 and 8 ng/μL, respectively (Appendix A—Table A6). The LOD was 5 and 8 ng/μL for PAPM and L2, respectively, determined after testing the preliminary screening concentration and the immediately higher (8 ng/μL) and lower (1.25 ng/μL) concentrations (Appendix A—Table A7). These values are much lower than the recommended DNA working concentration (20 ng/μL DNA), indicating that the method is sensitive in these conditions.

#### 3.5.5. Robustness

The standard deviation of the method (S) does not exceed the acceptance criteria in each of the tests performed (Appendix A—Table A8); therefore, the multi-locus PCR-HRM method is considered robust carried out under the conditions previously described.

#### 3.5.6. Transferability Test

Cohen’s kappa coefficient was 0.925, indicating an “almost perfect agreement” in the results of both laboratories (Landis and Koch, 1977). In the 220 samples analyzed, only 15 individuals (6.8%) did not show agreement in the species identification between both laboratories (Appendix A—Table A9). The non-matching results arise because, in one laboratory, individuals showed two or more non genotyped loci, resulting in a GeneClass2 score under 95%, not allowing to assign the species. No individuals were assigned to different species by the participant laboratories.

## 4. Discussion

We developed a multi-locus assay for species identification in the *Mytilus* genus based on the 10 most informative SNPs from 123 analyzed. Using the multi-locus genotypes, the four target species, Chilean mussel (*M. chilensis*), Blue mussel (*M. edulis*), Mediterranean mussel *(M. galloprovincialis)* and the Bay mussel (*M. trossulus*), can be confidently identified. This method allowed us to discard the presence of other mussels not belonging to the *Mytilus* genus but commonly found in seafood products, such as *A. atra* and *C. chorus.*

### 4.1. Multi-Locus PCR-HRM Method Development

The *F**_ST_* outlier and MAF_MAX_ criteria were useful to select the most informative SNPs for SI that are intended to reveal variations among but not within species. The reduced 10 SNP panel was constructed including the SNPs selected by each of these two criteria.

The MAF_MAX_ criteria maximize the difference in allelic frequencies among species, selecting locus showing a maximum allelic frequency for one species, but a minimum for another (MAX_MAF_) [36]. However, to use this approach it is necessary to sample several locations per species and with enough number of individuals, to rule out that the observed allele frequencies are population specific. This criterion is similar to the one used by Wilson et al. [30], these authors choose the more informative SNP using the “loading values” calculated after a discriminant analysis of principal components. Higher loading values reflects SNPs that capture more variance in the allele frequencies and therefore greater differences among species.

A locus shows high *F**_ST_* values when two or more groups are homozygous for contrasting alleles. Therefore, the *F**_ST_* outlier criterion has been used to select SNPs for geographic assignment because differences in their allele frequencies are probably a product of local adaptation [12,63]. In the *Mytilus* genus, species are restricted to specific geographic areas, especially the native species from the southern hemisphere, therefore, *F**_ST_* outlier SNPs are expected to be informative to differentiate them.

Wilkinson et al. [64] have shown a low correlation between both criteria selecting informative loci, a fact corroborated by our study. Only two loci (PAPM and L7) were selected by MAF_MAX_ and *F**_ST_* outlier at the same time. Therefore, it is important to considerer both criteria together when selecting the most informative loci for SI.

Searching for loci that can be technically genotyped, each of the potentially most informative SNPs must be revised to determine the feasibility to design primers. Moreover, to obtain clearly distinguishable melting curves, amplicons containing more than one polymorphism should be discarded. Other technical aspects of HRM analysis must be also considered [65]. However, optimal conditions scarcely happened, and to design a practical assay it is necessary to be flexible. For example, we included one SNP A/T because was informative to separate between *M. edulis* and *M. galloprovincialis* although these kinds of SNPs are not recommended as they are hard to genotype. We also removed some C/T and G/A SNPs because they gave redundant information.

The performance of the reduced 10 SNP panel was similar to the obtained with 49 SNPs (Dataset A) and 90 SNPs (Dataset B). Assignment probability, sensitivity and specificity were maximum for all four species tested here, indicating that the method performs well, correctly identifying the species and excluding individuals that do not correspond to species false positive. To our knowledge, the only reduced multi-locus panel to perform SI in the three European species was developed by Wilson et al. (2018) [30]. The performance of this 12 SNP panel, calculated from their raw data and excluding populations used for validation purposes, showed a maximum specificity for the three species. Sensitivity assigning *M. trossulus* and *M. galloprovincialis* was also a maximum; however, in *M. edulis*, this value reaches only 0.96. The panel developed here can detect one species more with fewer SNPs (10 vs. 12) and with higher performance. As Wilson et al. (2018) [30] have clearly shown, the most valuable aspect of multi-locus panels is the identification of putatively hybrid not detected by the mono-locus assays.

### 4.2. Fitness for Purpose Assessment of the Multi-Locus PCR-HRM Method

Before the commercial application of any analytical method, it must be validated following accepted quality criteria [37,38]. In laboratories involved in food testing, compliance with ISO/IEC Standard 17025 [40] requirements is mandatory. However, formal validation studies of qualitative techniques that use the PCR-HRM technique to identify marine species are scarce in the current literature. In the case of qualitative PCR-based methods, their scope or applicability relies on the capacity to obtain enough DNA with proper integrity and purity, and also in the absence of PCR interferents. In our study, the CTAB method allowed us to extract enough quality DNA in all tested food matrices. However, every testing laboratory must validate its own DNA extraction method before using this multi-locus assay.

The canning process involves the application of high temperatures (≈121 °C) to the product and its packing media, hermetically sealed in an anaerobic environment [66]. Is widely recognize that in highly processed food (canned or cooked), the DNA is fragmented below 300 bp, leading to extracting less and degraded DNA [9,67]. The HRM analysis use amplicons shorter than 300 bp [11] and the sensitivity of detection is enhanced in smaller amplicons [65]; therefore, it is ideal for genetic analysis of highly processed food. In our method, the size of the amplicons sizes were between 50 to 170 bp, appropriated to be successfully genotyped in processed food. However, the packing media may contain organic acids, ions, chelating agents and other ingredients that favor DNA fragmentation during the thermal process and/or interfere with the PCR reaction [67,68,69,70]. Although we obtained DNA of high quality from products packed in vinegar or sauces containing vinegar, amplification failed or the *Cq* values were larger than the threshold defined to obtain genotypes. It has been demonstrated that using commercial silica-based columns along with chaotropic solutions allow the removal of some PCR inhibitors in comparison with non-commercial methods such as the phenol–chloroform [71]. Our results pose the challenge to further testing to remove these interferents and/or using amplification enhancers [72,73], to expand the scope of our assay to include complex matrices.

The method was practical and easy to apply, and requires short training time. The HRM technology works in a closed tube, reducing human effort, errors and the risk of drag contamination [74,75]. In addition, it is amendable with small laboratories with low samples flow. The reagents cost are affordable (≈USD 23 per sample in Chile) and could be reduced in countries where it is not necessary to import reagents and consumables.

Although three primer pairs retrieve potential in silico amplifications during Primer Blast analysis, and two of them showed amplification during the in vitro assays in non-target species, the method was specific. The multi-locus approach means that a specimen is assigned to the species using the information of the 10 loci simultaneously. Therefore, the isolated amplification of one or two SNPs in the non-target species will not produce any species assignment. This panel avoid assigned errors derived from the analysis performed in a mono-locus way.

Although this method is qualitative, the DNA concentration that can be detected with 95% confidence (LOD_95%_ = 5 and 8 ng/μL) was 25–40% of the DNA concentration recommended in the SOP of analysis (20 ng/μL), showing adequate sensitivity for the intended use. The LOD of this qualitative method was appropriated on the base of the *Cq* values ranges, all positive and specific amplification observed showed *Cq* values below 29 [38,65]. The method was robust and stable in front of small variations in the eight common operational parameters tested in an orthogonal test as recommended by ISO 20813:2019 [76]. Transferability to a second laboratory allowed the evaluation of the method under other different condition (operator, equipment, etc.). As the method is multi-locus, when one lab fails to amplify some loci, the SI of the sample cannot be obtained. Therefore, no discrepancies in species assignment arise.

## 5. Conclusions

The multi-locus PCR-HRM method developed in this study was applicable to identify *M. chilensis*, *M. edulis*, *M. galloprovincialis* and *M. trossulus* mussels, fresh, frozen and canned with brine, oil and scallop sauce. As a limitation, the method is not 100% applicable or efficient in preserves that contain acetic acid (wine vinegar) and tomato sauce. The method is a practical, fast, sensitive and robust diagnostic tool to differentiate the target species of the genus *Mytilus* from others commonly found in seafood containing mussel.

## Figures and Tables

**Figure 1 foods-10-01684-f001:**
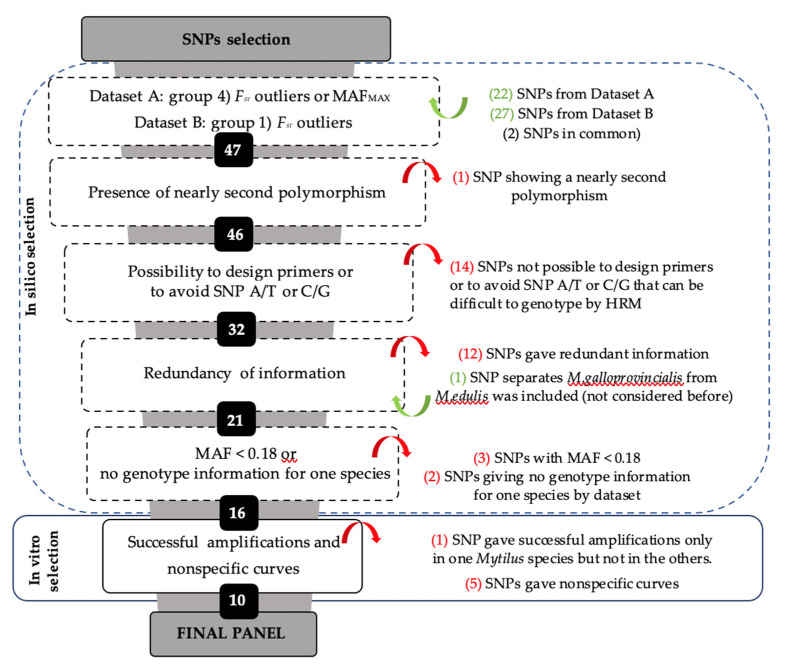
Workflow for the selection of a reduced single nucleotide polymorphism (SNP) panel for species identification in *Mytilus* genus. Each box represents a stage with the remaining number of SNPs. Filtering criteria were summarized on the right side of each stage. In green is the number of loci added; in red is the number of loci removed. Minor allele frequency (MAF).

**Figure 2 foods-10-01684-f002:**
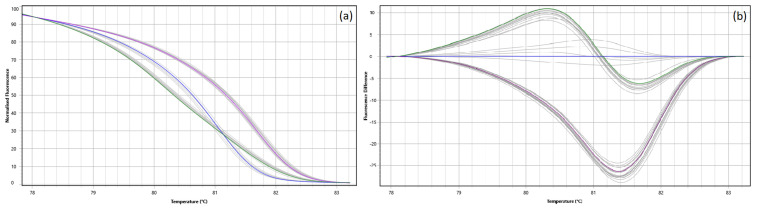
Example of PCR-HRM curves for the L7 SNP with alleles C/T. Each curve corresponds to one of the three possible genotypes. In purple (**―**) control curve C/C, in blue (**―**) control curve (T/T), in green (**―**) control curve (C/T) and in grey (―) query samples. (**a**) Normalized melting curves; (**b**) difference melting curves, using control curve for genotype T/T as reference.

**Table 1 foods-10-01684-t001:** Experimental design including type and number of samples analyzed in each step of the develop and validation stages.

Species	*n*	Country of Origin	Product	Packing Media	Development	Validation				
					D	A	P	LOD	R	T
Natural Populations										
*Mytilus chilensis*	34	Chile	Raw	-	X	X	X	X	X	X
*Mytilus edulis*	34	Canada	Raw	-	X	X	X	X	X	X
*Mytilus galloprovincialis*	34	Spain	Raw	-	X	X	X	X	X	X
*Mytilus trossulus*	34	Canada	Raw	-	X	X	X	X	X	X
*Aulacomya atra*	20	Chile	Raw	-	X	-	X	-	-	X
*Choromytilus chorus*	20	Chile	Raw	-	X	-	-	-	-	X
*Perumytilus purpuratus*	20	Chile	Raw	-	X	-	-	-	-	-
Commercial Samples										
ND	3	Chile	Canned	Brine	-	X	-	-	-	-
*Mytilus edulis*	3	Germany	Canned	Brine	-	X	-	-	-	-
ND	3	Spain	Canned	Brine	-	X	-	-	-	-
ND	3	Chile	Canned	Oil	-	X	-	-	-	-
ND	3	Chile	Canned	Oil	-	X	-	-	-	-
ND	3	Chile	Canned	Marinated	-	X	-	-	-	-
*Mytilus edulis*	3	Germany	Canned	Marinated	-	X	-	-	-	-
ND	3	Spain	Canned	Marinated	-	X	-	-	-	-
ND	3	Chile	Canned	Hot sauce	-	X	-	-	-	-
ND	3	Spain	Canned	Scallop sauce	-	X	-	-	-	-
*Mytilus edulis*	3	Denmark	Canned	Tomato sauce	-	X	-	-	-	-
ND	3	Chile	Frozen	Pre-Cooked	-	X	-	-	-	-

Abbreviations: A, applicability; D, diagnostic test performance (sensitivity and specificity of the method) and primer specificity; LOD, limit of detection; P, practicability; R, robustness; S, SNP selection; T, transferability test. X, Samples used to perform this step; ND: Species not declared.

**Table 2 foods-10-01684-t002:** Parameters used in eight trials for the robustness study *.

		Measurement Level (High–Low)							
Factor	Standard Value	1	2	3	4	5	6	7	8
Annealing temperature (°C)	59	60	60	60	60	58	58	58	58
260/230 ratio	≈2.0–2.2	2.1	2.1	0.7	0.7	2.1	2.1	0.7	0.7
260/280 ratio	≈1.8–2.0	2.0	1.7	2.0	1.7	2.0	1.7	2.0	1.7
Annealing time (s)	30	40	40	20	20	20	20	40	40
Primer concentration (ng/μL)	10	15	5	15	5	5	15	5	15
Reaction volume (μL)	8	9	7	7	9	9	7	7	9
Reagent (HRM Kit)	-	S	Q	Q	S	Q	S	S	Q

HRM (High Resolution Melting) Kit: S (SensiFast™ HRM Kit, Meridian Biociences^®^) and Q (qPCRBIO™ HRM Mix, PCRBiosystems). * Based on Youden and Steiner (1975) [60].

**Table 3 foods-10-01684-t003:** Assignment success in datasets A and B by criteria.

Group	Dataset A (*n* = 311)		Dataset B (*n* = 386)	
	N° of SNP	Correctly Assigned Individuals (%)	N° of SNP	Correctly Assigned Individuals (%)
All SNP	49	309 (99.4)	90	363 (94.0)
(1) *F_ST_* outliers	11	287 (92.3)	27	363 (94.0)
(2) MAF_MAX_	17	308 (99.0)	21	358 (92.7)
(3) *F_ST_* outliers and MAF_MAX_	6	286 (92.0)	11	354 (91.7)
(4) *F_ST_* outliers or MAF_MAX_	22	308 (99.0)	37	359 (93.0)

**Table 4 foods-10-01684-t004:** Species identification in commercial samples.

Commercial Samples	Packed in	Declared Species	*n*	Species Identified				
				Mch	Me	Mg	Mt	NA
Canned in brine	Chile	ND	3	3				
Canned in brine	Germany	*M. edulis*	3		3			
Canned in brine	Spain	ND	3			3		
Canned oil	Chile	ND	3	2				1
Canned oil	Chile	ND	3	3				
Canned marinated	Chile	ND	3					3
Canned marinated	Germany	*M. edulis*	3	1				2
Canned marinated	Spain	ND	3					3
Canned hot sauce	Chile	ND	3	3				
Canned scallop sauce	Spain	ND	3			2		1
Canned tomato sauce	Denmark	*M. edulis*	3					3
Frozen pre-cooked	Chile	ND	3	3				

*n*: Number of individuals analyzed. ND: Species not declared. NA: Not assigned species. Mch: *M. chilensis.* Me: *M. edulis.* Mg: *M. galloprovincialis.* Mt: *M. trossulus.*

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
