# Peer review of "Development and Validation of a Multi-Locus PCR-HRM Method for Species Identification in Mytilus Genus with Food Authenticity Purposes"

_foods, 2021, doi:10.3390/foods10081684_

Round 1

Reviewer 1 Report

The manuscript “Development and Validation of a Multi-Locus PCR-HRM Methodd for Species Identification in Mytilus Genus with Food Authenticity Purposes” seems to address an important issue of food authentication. The manuscript is well-written, although typos can be found along the manuscript, which need to be corrected (including in the title). The manuscript could be of potential interest for the regulatory agencies and governmental food analysis laboratories, therefore it has potential after addressing the following comments.

Comments:

Title - correct typos. The genus name is in italic.

Abstract – LOD6 is explained latter but its reference in the abstract without context does not make any sense. Additionally, this designation should be revised to a more commonly known nomenclature. Please revise according to this comment and comment in line

Line 49 - correct to "... aimed at confirming the authenticity of the products"

Line 49 - What do the authors mean with "... and protect them from global competition."? Please clarify.

Line 93 - What do the authors mean with "... in-silico and in-vitro stages..." Please clarify.

Table 1 - what were the diagnostic tests used? This table needs further information. Please revise.

Lines 155-156 - Did the authors check for the presence of RNA content? Contamination with RNA can influence DNA amplification, thus destabilising the extracts. Did the authors add RNase to their extraction protocols? Please explain and detail.

Line 156 - correct unit to L. Check for similar situations along the manuscript.

Lines 162-163 - 0.1ºC per second seems rather high temperature increments for adequate HRM analysis, since rapid transitions may difficult melting curve analysis. Please explain the use of this protocol.

Lines 163-165 - Please, provide the name of the primers and the size of PCR products.

2.3. PCR-HRM Analysis and primer design section – Please detailed which were the parameters used for HRM analysis.

Lines 216-220 - LOD can be expressed either as absolute or relative, when assessing serial diluted single species DNA or DNA of a species within a matrix. The LOD is always the lowest concentration level that amplifies at least 95% of the times. Therefore, the designations attributed to LOD by the authors do not make much sense. Please change to a more adequate designation.

Figure 2. - the authors should add to fig 2, the difference curves corresponding to Fig 2 graph.

Line 322 - RT-PCR corresponds to reverse transcriptase PCR. Please correct to real-time PCR.

Line 324 - USD 10 is the cost per sample analysis? Because this is not clear in the sentence. Please clarify.

Please, provide an example of HRM analysis with different references species and commercial samples.

References – some references are not well formatted.

Author Response

Thank you very much for your careful revision and improvement suggestions. 

1. Title: correct typos.
Response: The typos were corrected, in the title and along the document.
2. Abstract – LOD6 is explained latter but its reference in the abstract without context does not make 
any sense. Additionally, this designation should be revised to a more commonly known nomenclature. 
Please revise according to this comment and comment in line.
Response: We change the phrase in the abstract, as follows: “Sensitivity, expressed as limit of 
detection (LOD) ranged between 5 and 8 ng/μL.”.
3. Line 49 - correct to "... aimed at confirming the authenticity of the products" 
Response: Done.
4. Line 49 - What do the authors mean with "... and protect them from global competition."? Please 
clarify." 
Response: This paragraph was deleted.
5. Line 93 - What do the authors mean with "... in-silico and in- vitro stages..." Please clarify.
Response: We agree that it was confusing, so we simplify the phrase erasing “in-silico” and “in- vitro”
in this context.
6. Table 1 - what were the diagnostic tests used? This table needs further information. Please revise." 
Response: The experimental design table and its title and foot notes, were reviewed and reorder to 
clarify. The information about the diagnostic tests used were added as foot notes.
7. Lines 155-156 - Did the authors check for the presence of RNA content? Contamination with RNA 
can influence DNA amplification, thus destabilizing the extracts. Did the authors add RNase to their 
extraction protocols? Please explain and detail.
Response: We added in the MS the phrase (Line 191): “Methods i) and ii) include a RNAse 
treatment, however, as a rapid method, iii) Chelex® 100 method does not.”
The Phenol-chloroform method modified for Mytilus [45] use 200 µg of RNAse and the ii) Cetyl 
trimethyl ammonium bromide (CTAB) modified by Arseneau et al, [46] added 50 µg of RNAseA
45. Larraín, M.A.; Díaz, N.F.; Lamas, C.; Vargas, C.; Araneda, C. Genetic composition of Mytilus species in 
mussel populations from southern Chile. Lat. Am. J. Aquat. Res. 2012, 40, 1077–1084, doi:10.3856/vol40-
issue4-fulltext-23.
46. Arseneau, J.-R.; Steeves, R.; Laflamme, M. Modified low-salt CTAB extraction of high-quality DNA from 
contaminant-rich tissues. Mol. Ecol. Resour. 2017, 17, 686–693, doi:https://doi.org/10.1111/1755-0998.12616.
8. Line 156 - correct unit to L. Check for similar situations along the manuscript.
Response: We correct this typo across the MS.
9. Lines 162-163 - 0.1ºC per second seems rather high temperature increments for adequate HRM 
analysis, since rapid transitions may difficult melting curve analysis. Please explain the use of this 
protocol. 
Response: The Real time PCR equipment used (ECO™ Real-Time PCR System (Illumina) and a 
MIC qPCR Cycler (BioMolecular System) allowed a minimum increment of 0.1 ºC/sec. This 
temperature increment was enough to genotype the locus L2 bearing an A/T SNP without any 
problems. 
10. Lines 163-165 - Please, provide the name of the primers and the size of PCR products. 
Response: We added a supplementary table (Appendix A – Table A2) with this information and the 
polymorphism in each locus.
11. 2.3. PCR-HRM Analysis and primer design section – Please detailed which were the parameters 
used for HRM analysis.Response: The parameters used for HRM analysis, Temperature, times, reagents and reaction mix 
composition were described in section 2.3. Which extra parameters are necessary to include?. Please 
let us know.
12. Lines 216-220 - LOD can be expressed either as absolute or relative, when assessing serial 
diluted single species DNA or DNA of a species within a matrix. The LOD is always the lowest 
concentration level that amplifies at least 95% of the times. Therefore, the designations attributed to 
LOD by the authors do not make much sense. Please change to a more adequate designation.
Response: Initially we used the LOD definition in Broeders et al, 2014. As the reviewer suggested, 
we change to a more universal definition of LOD. Consequently, this section was extensively modify 
in the text. 
Broeders, S.; Huber, I.; Grohmann, L.; Berben, G.; Taverniers, I.; Mazzara, M.; Roosens, N.; Morisset, D. 
Guidelines for validation of qualitative real-time PCR methods. Trends Food Sci. Technol. 2014, 37, 115–126, 
doi:10.1016/j.tifs.2014.03.008.
13. Figure 2. - the authors should add to fig 2, the difference curves corresponding to Fig 2 graph.
Response: We added the difference curves to figure 2.
14. Line 322 - RT-PCR corresponds to reverse transcriptase PCR. Please correct to real-time PCR.
Response: We change RT-PCR to real-time PCR along the text.
15. USD 10 is the cost per sample analysis? Because this is not clear in the sentence. Please clarify.
Response: We clarify that this cost is per sample and change the cost to USD 23 because we also 
considered the standards and negative controls. 
16. Please, provide an example of HRM analysis with different references species and commercial 
samples.
Response: As example we add a table (Table 4) with the results of the SI in the commercial samples
per food matrix. Also, the following paragraph was added in the text: “An example of the results of SI 
in the commercial samples used for the applicability step is presented in Table 4. This table are 
shown only samples with eight or more SNP successfully genotyped, allowing SI. Species was 
identified in frozen products and packed in brine, oil, hot and scallop sauces. In products packed 
marinated or in tomatoes sauce, the lower number of SNPs genotyped hampered SI. Although 
canned marinated products are not within the scope of the method, it is interesting to note that, the 
only species that could be identified in this matrix, does not match with the one declared in the label 
(Table 4)”.

Reviewer 2 Report

The manuscript of Quintrel et al. proposes a Multi-locus PCR-HRM approach for the identification at species level of Mytilus genus. The topic is interesting, given that authentication of mussels becomes difficult especially when the external morphological characteristics are removed, and the identification of specimens from hybrid zone is could be difficult by mono-locus approach.

Overall, the paper has been well structured and I have few comments for authors.

  1. Title: Remove double “d” from “Method” and write Mytilus in Italics
  2. Abstract: Write Mytilus and all names of mussels in Italics
  3. Introduction: Lines 50-51. Better explain all possible implications in seafood mislabelling (i.e. sustainability, ect…)
  4. Materials and methods:

- for a better understanding, please pose sample collection, DNA extraction and species identification in separate paragraphs

- Authors explained in the discussions that “packing media may contain organic acids,…that could inhibit PCR”. Also they affirmed that interferents or amplification enhancers could be used to maximize remove limits occurred in the identification of preserves that contain acetic acid and tomato sauce.

Were “negative” PCR reactions subjected to spiked controls for identification the presence of PCR inhibitors, as documented in the literature?

  1. Table 1. Please, add in legend the meaning of “P”
  2. Line 276: Change 3.4.1 with 3.5.1
  3. Reference section: should be updated with more recent references

Author Response

Thank you very much for your positive and helpful comments.

1. Title: Remove double “d” from “Method” and write Mytilus in Italics
Response: Done.
2. Abstract: Write Mytilus and all names of mussel in Italics
Response: Done.
3. Introduction: Lines 50-51. Better explain all possible implications in seafood mislabeling (i.e. 
sustainability, ect...)
Response: Done. We add in the text, a brief explanation about the possible implications mentioned
as follows:
“Species substitution can result in an inexpensive product being labelled as high-priced, but can also 
affects food safety by unnoticed consumption of allergens due to undeclared species [7,8]. Seafood 
mislabeling is well documented throughout history [9,10], it impacts not only food authenticity [11], 
also allows the trade in the markets of endangered species or products from illegal, unreported, and 
unregulated (IUU) fisheries, threatening wildlife [12], hampering conservation and negatively affecting 
consumers rights [13]”.
4.Materials and methods: 
- for a better understanding, please pose sample collection, DNA extraction and species identification 
in separate paragraphs Response: We separate the information in two sections: 2.2 Sample collection and 2.3 DNA 
extraction. 
4a. Materials and methods: 
- Authors explained in the discussions that “packing media may contain organic acids,...that could 
inhibit PCR”. Also, they affirmed that interferents or amplification enhancers could be used to 
maximize remove limits occurred in the identification of preserves that contain acetic acid and tomato 
sauce. Were “negative” PCR reactions subjected to spiked controls for identification the presence of 
PCR inhibitors, as documented in the literature? 
Response: No, in the case of negative PCR reactions, we did not check for the presence of PCR 
inhibitors. This paper focus in the development and validation of the method, we do not seek to solve 
the problem of possible interferers to broaden the scope of the method here. The PCR inhibitors is a 
general and very relevant problem for any PCR based methods, that we plan to address in future 
works.
5.Table1.Please, add in legend the meaning of “P” 
Response: Done. The information about “P” were added as foot notes.
6.Line 276: Change3.4.1 with 3.5.1 
Response: Done.
7.Reference section: should be updated with more recent references
Response: We revise the recent methods for Si in Mytilus published, and added the following work : 
del Rio-Lavín, A.; Jiménez, E.; Pardo, M.Á. SYBR-Green real-time PCR assay with melting curve analysis for the 
rapid identification of Mytilus species in food samples. Food Control 2021, 130, 108257.

Reviewer 3 Report

I am sending you the revision of the manuscript entitled “Development and Validation of a Multi-Locus PCR-HRM Method for Species Identification in Mytilus Genus with Food Authenticity Purposes”. The authors developed a multi-locus assay for species identification in the Mytilus genus based on ten informative SNPs selected throughout a methodological approach.

Overall, the paper is clear and well-written. The methodology is carefully described, and each analytical step is detailed, supported, and performed with rigorous and robust methods. I especially appreciated the authors’ choice to include the sections “applicability” and “practicability”, which concretely allow to consider the use of this method at inter-laboratories level.

Main aspects

  • I suggest the authors include the results from the commercial samples PCR-HRM analysis. I think it would be interesting to provide information on the market status of these products and the used species, especially considering the different origin of the samples (Chile, Germany, Spain and Denmark).
  • Although the authors stated that the method allow to identify hybrids, no detailed information on this aspect was provided. It would be useful to include a figure on PCR-HRM curves using hybrid specimens for observing the difference with pure specimens.

Revision

Title: Modify “methodd” in “method”

Lines 26-28: Please write the species name in italics

Line 29: Please briefly clarify the meaning of LOD5 and LOD95 also in the abstract.

Line 37: maybe the authors intended that mussel represent 12% of the worldwide seafood production.

Line 38: Did the authors intend bivalve or mussels? If they intended bivalves, please modify in the singular.

Lines 42-43: Pleas add “together with other Mytilus spp.” after “World Register of Marine species”.

Line 63: Jilberto et al.

Line 105: The first criterion selected (in the singular). Modify in the singular also in line 111.

Lines 275-276: the section “applicability” is 3.5.1 instead of 3.4.1

Author Response

Thank you very much for your work and improvements.

Main aspects
1. I suggest the authors include the results from the commercial samples PCR-HRM analysis. I think it 
would be interesting to provide information on the market status of these products and the used 
species, especially considering the different origin of the samples (Chile, Germany, Spain and 
Denmark).
Response: Done, we added Table 4 containing the species identification results using the proposed 
multi-locus method in commercial products. In the text we added the following paragraph: ”An 
example of the results of SI in the commercial samples used for the applicability step is presented in 
Table 4. This table are shown only samples with eight or more SNP successfully genotyped, allowing 
SI. Species was identified in frozen products and packed in brine, oil, hot and scallop sauces. In 
products packed marinated or in tomatoes sauce, the lower number of SNPs genotyped hampered SI. 
Although canned marinated products are not within the scope of the method, it is interesting to note 
that, the only species that could be identified in this matrix, does not match with the one declared in 
the label (Table 4).
2. Although the authors stated that the method allow to identify hybrids, no detailed information on this 
aspect was provided. 
Response: We are aware that a procedure to identify hybrids is needed, we will address this topic in 
another paper using simulated hybrids because the hybrid frequency of M. chilensis x M. 
galloprovincialis in Chilean coast is low (4-7%) (Larraín et al. 2019). Theoretically, in hybrid detection, 
our method is equivalent to Wilson et al (2018).
Larraín, M. A., González, P., Pérez, C., & Araneda, C. (2019). Comparison between single and multi-locus 
approaches for specimen identification in Mytilus mussels. Scientific Reports, 9. https://doi.org/10.1038/s41598-
019-55855-8
Wilson, J.; Matejusova, I.; McIntosh, R.E.; Carboni, S.; Bekaert, M. New diagnostic SNP molecular markers for the 
Mytilus species complex. PLoS One 2018, 13, e0200654, doi:10.1371/journal.pone.0200654.
3. It would be useful to include a figure on PCR-HRM curves using hybrid specimens for observing 
the difference with pure specimens. Response: In Figure 1, the heterozygous individual (in green) is an hybrid individual between M. 
chilensis (allele C) and M. galloprovincialis (allele T) for the SNP L7.
Revision 
4. Title: Modify “methodd” in “method”
Response: Done.
5. Lines 26-28: Please write the species name in italics 
Response: Done.
6. Line 29: Please briefly clarify the meaning of LOD5 and LOD95 also in the abstract. 
Response: Done. The section 2.6.4 Sensitivity (LOD) was completely rewritten to use the more 
known terminology of Limit of detection in all the paper, including the abstract.
7. Line 37: maybe the authors intended that mussel represent 12% of the worldwide seafood 
production. 
Response: We change the phrase to state that “Mussels from Mytilus genus represented 20% of the 
worldwide mollusks production in 2018”. 
8. Line 38: Did the authors intend bivalve or mussels? If they intended bivalves, please modify in the 
singular. 
Response: Done. Bivalves was modified to singular.
9. Lines 42-43: Pleas add “together with other Mytilus spp.” after “World Register of Marine species”. 
Response: Done. 
10. Line 63: Jilberto et al. 
Response: Done.
11. Line 105: The first criterion selected (in the singular). Modify in the singular also in line 111. 
Response: Done, we made three changes to the singular. 
12. Lines 275-276: the section “applicability” is 3.5.1 instead of 3.4.1
Response: Done.

Reviewer 4 Report

The paper describes development and validation of a PCR-HRM test based on the analysis of 10 highly conservative SNPs to identify species belonging to Mytilus genus. The research design is appropriate and the validation is well conducted. The results demonstrated the suitability of the method for the species identification of several kind of samples (fresh, frozen and canned with brine, oil and scallop sauce), but low efficiency on preserves containing acetic acid and tomato sauce has been proven. In my opinion it would be important to improve the test efficiency on these matrices by using commercial silica based kits for DNA extraction. 

Some revisions to the text are needed:

  • Title: please correct the word "method"
  • In the methods description primer sequences are missing.
  • Line 174-5: subject of the sequence is lacking
  • line 204-5:please rewrite the sentence,  subject is lacking
  • line 276: the applicability section is identified by 3.5.1 not 3.4.1. as indicated
  • table 4 is not necessary
  • line 405: please remove the bracket after "false positive"

Author Response

Thank you very much for your useful comments and detailed revision.

1. The results demonstrated the suitability of the method for the species identification of several kind 
of samples (fresh, frozen and canned with brine, oil and scallop sauce), but low efficiency on 
preserves containing acetic acid and tomato sauce has been proven. In my opinion it would be 
important to improve the test efficiency on these matrices by using commercial silica based kits for 
DNA extraction. 
Response: We agree with this comment, and we are aware that for the commercial use of the 
method is important, to broaden its scope to include such matrices. However, the present MS is 
focused in the development and validation of the method, we do not seek to address this limitation in 
this paper.
2. Title: please correct the word "method" 
Response: Done.
3. In the methods description primer sequences are missing. 
Response: This method will be protected with a patent, therefore primer sequences are not publicly 
available at the moment. However we add some information about the primers that not compromise 
the protection process in Appendix A-Table A2.
4. Line 174-5: subject of the sequence is lacking 
Response: Done. The subject was added. “It was …”.
5. line 204-5:please rewrite the sentence, subject is lacking
Response: Done. The subject was added. “It was …”.6. Line 276: the applicability section is identified by 3.5.1 not 3.4.1. as indicated 
Response: Done.
7. table 4 is not necessary
Response: Done. Table 4 was moved to the Appendix A – Table A3
8. line 405: please remove the bracket after "false positive"
Response: Done